# Development and Assessment of Duplex and Triplex Laminated Edible Films Using Whey Protein Isolate, Gelatin and Sodium Alginate

**DOI:** 10.3390/ijms21072486

**Published:** 2020-04-03

**Authors:** Andrey A. Tyuftin, Lizhe Wang, Mark A.E. Auty, Joe P. Kerry

**Affiliations:** 1Food Packaging Group, School of Food and Nutritional Sciences, University College Cork, Western Road, Cork T12 K8AF, Ireland; a.tiuftin@ucc.ie (A.A.T.);; 2Reading Scientific Services Ltd., Reading Science Centre, Whiteknights Campus, Pepper Lane, Reading, Berkshire RG6 6LA, UK

**Keywords:** compostable, edible films, duplex, triplex laminates, whey protein, gelatin, sodium alginate, film structure, mechanical, barrier properties

## Abstract

The objective of this study was to assess the ability of producing laminated edible films manufactured using the following proteins; gelatin (G), whey protein isolate (WPI) and polysaccharide sodium alginate (SA), and to evaluate their physical properties. Additionally, films’ preparation employing these ingredients was optimized through the addition of corn oil (O). Overall, 8-types of laminated films (G-SA, G-WPI, SA-WPI, SA-G-WPI, GO-SAO, GO-WPIO, SAO-WPIO and SAO-GO-WPIO) were developed in this study. The properties of the prepared films were characterized through the measurement of tensile strength (TS), elongation at break point (EB), puncture resistance (PR), tear strength (TT), water vapour permeability (WVP) and oxygen permeability (OP). The microstructure of cross-sections of laminated films was investigated by scanning electron microscopy (SEM). Mechanical properties of films were dramatically enhanced through the addition of film layers. GO-SAO laminate showed the best barrier properties to water vapour (22.6 ± 4.04 g mm/kPa d m^2^) and oxygen (18.2 ± 8.70 cm^3^ mm/kPa d m^2^). SAO-GO-WPIO laminate film was the strongest of all laminated films tested, having the highest TS of 55.77 MPa, PR of 41.36 N and TT of 27.32 N. SA-G-WPI film possessed the highest elasticity with an EB value of 17.4%.

## 1. Introduction

Conventional packaging materials, which have included the use of wood, paper-based products, glass, metals (primarily steel and aluminium) and plastics, plastic-based laminates and co-extrusions have formed the cornerstone to those formats which have, and still for the most-part, underpin the commercial packaging used around retailed products and throughout all three packaging levels, primary (sales), secondary (collation and handling) and tertiary (transport). Flexible packaging materials such as, for example, oil-sourced laminates (including, but not limited to polyethylene terephthalate, polyethylene and polyamide) have led to the reduction of product costs (especially for food), wide and almost constant availability of product types regardless of seasonality, creation of competition in the marketplace, and provision of consumer convenience in many different forms (large and small pouches, flow packs and others). This is particularly true for food and beverage products, where features of product safety, hygiene and shelf-life stability have become important as features of convenience, particularly through packaging’s role in reducing food waste. When discussing the problems associated with food packaging materials on environmental grounds, it is extremely important not to lose sight of the role played by packaging to date in reducing food wastage, surely an issue as important as packaging waste, if not more so. Therefore, while much controversy surrounds the use of current, conventional, non-biodegradable packaging materials such as plastics and plastic-based packaging on environmental grounds, it is important that we do not exert ’knee-jerk’ reactions, which result in adopting packaging materials and formats, which are not robust enough to successfully contain, protect or preserve food products. This will only result in creating further negative environmental issues. Packaging pollution of our global environment has been in the making since the 1950s and the real failure associated with packaging is not the packaging waste itself, but the human failure in the first instance to put systems in place to deal with different packaging materials when initially introduced into the marketplace for use.

Biodegradable and compostable materials, which can fully degrade in soil or the ocean, are becoming a solution for the environmental issue of plastic packaging wastes currently. Thus, changing of the molecular structure in laminates made of conversional packaging materials to the structures containing biodegradable and compostable natural sourced polymer chains can lead to saving of the environment.

There is increased interest in the manufacture of compostable edible films from proteins, polysaccharides and lipids [1,2,3,4,5]. Some research efforts have focused on attempting to improve the properties of compostable films by laminating additional film layers [6,7]. Triplex laminated films containing plied dialdehyde starch, cross-linked gelatin (outer layer) and plasticized gelatin with sodium montmorillonite film (inner layer) were studied by [8]. The same authors found that the use of additional film layers increased the most desired properties found in a monolayer gelatin film [8]. Different compositions of whey protein isolate, hydrolysed whey protein isolate and glycerol were studied [9]. Bilayer films based on methylcellulose, whey protein, wheat gluten or zein laminated with a lipid mixture of fish gelatin and emulsified gelatin bilayer films have been previously studied including composites with chitosan [5,9,10,11,12,13]. Whey protein isolate and gelatin composite films have also been studied by [13]. Chitosan/gelatin bilayer composite films have been shown to improve mechanical, transport and physical film properties compared to monolayer films [14]. In most cases, water vapour permeability of monolayer films was improved by their lamination with other film types. Overall, gas barrier properties of monolayer layer films can be improved through the lamination and using film formulations containing fatty acids [1,10,15,16,17,18]. For example, it has been reported that acceptable potato chip quality was maintained for up to 43 days at 50% RH using laminated methylcellulose/corn zein edible films with stearic and palmitic acids added to the corn zein layer [18].

The manufacture of multilayer films that may be edible/biodegradable/compostable in nature are of a great interest to the food and packaging industries currently because of pressing environmental concerns surrounding the continued use of plastics and plastic-based laminates and the need to find packaging alternatives, which will deliver similar storage and shelf-life functions equivalent to synthetic conventional packaging forms. 

The development of multilayer, laminated, edible/biodegradable/compostable films produced from whey protein isolate (WPI), gelatin (G) and sodium alginate (SA) has received little attention. The optimal formulations of these ingredients in the formation of monolayer films have been reported previously [16,19,20,21,22]. Consequently, it seems logical and interesting to know if the properties of these single material films could be improved by lamination to each other.

Thus, the primary objectives of this study were to develop laminated edible/biodegradable/compostable films consisting of G, WPI and SA and to optimize films properties through the addition of corn oil (O) using a solvent casting technique and to assess their mechanical properties (tensile strength (TS), elongation at break (EB), puncture resistance (PR), tear strength (TT)), barrier properties (water vapour permeability (WVP) and oxygen permeability (OP)) and cross-sectional laminated structure by scanning electron microscopy (SEM).

## 2. Results and Discussion

### 2.1. Film Thickness

Laminated film thickness is an important feature in laminate development as thinner films are less perceivable to consumers, use less resources and, consequently, result in lower production costs. Therefore, preliminary trials were carried out to determine the least volume of film forming solutions required to produce a range of laminated edible/biodegradable/compostable films. In this study, the order in which film forming solutions were applied to create laminated structures was found to be important. For example, it was not possible to laminate G films onto WPI films due to the swelling, which occurred in WPI films when G solutions were cast onto WPI films. Overall, it was possible to create 8-types of laminated films using G, WPI and SA; six of which were bilayers and two types of which were trilayers (Table 1). The laminated films discussed in this study were all shown to peel easily from casting plates.

### 2.2. Mechanical Properties of Laminated Films

The mechanical properties of laminated films, in terms of tensile strength (TS), elongation at break point (EB), puncture resistance (PR) and tear strength (TT) are shown in Table 2. In general, means values for TS, EB, TT and PR as they pertained to trilayer films were higher than those determined for bilayer films. All laminated films that were produced using optimised formulations containing corn oil (O) had higher TS, E, TT and PT values compared to laminate films produced without the addition of O (Figure 1). 

The laminate SAO-GO-WPIO produced the strongest film among all tested laminate films in terms of tensile strength (55.8 ± 7.98 MPa), puncture strength (41.4 ± 10.01 N) and tear strength (27.3 ± 3.45 N). The laminated film SA-G-WPI had the highest elasticity with an EB value of 17.4% ± 0.03% (Table 2). No significant differences were determined between GO-WPIO and SAO-WPIO, G-WPI and GO-SAO, G-A and G-WPI in terms of TS, EB and TT, respectively. The remaining laminate possessed significant (*p* < 0.05) differences in terms of TS, EB, PR and TT values. Ranking films on the basis of decreasing tensile strengths showed that: SAO-GO-WPIO > GO-SAO > GO-WPIO = SAO-WPIO > G-SA > SA-G-WPI > SA-WPI > G-WPI, with TS values ranging from 55.77 N to 7.39 N. Ranking films by elongation at break point decreased in the following order: SA-G-WPI > G-SA > GO-WPIO = SAO-GO-WPIO > G-WPI = GO-SAO > SA-WPI > SAO-WPIO. The stiffest laminate film (SAO-WPIO) had an EB value of 5.0% ± 0.78%. There were no significant differences between G-WPI and GO-SAO, GO-WPIO and SAO-GO-WPIO. These results suggested that G was the predominant ingredient affecting the elongation of laminated films. Single layer G film had been shown to be the most elastic of all films when compared to WPI or SA monolayer films. Therefore, G film contributes the elongation attribute of laminate films when manufacturing multilayer films using WPI, SA and G ingredients [23]. From an extensive review of the scientific literature, no information appeared to be available on laminated films manufactured from these same ingredients. Ranking of PR values for all the laminate films decreased in the order: SAO-GO-WPIO > GO-SAO > SAO-WPIO > SA-G-WPI > GO-G-WPI > GO-WPI > GO-WPI > G-SA > G-WPI; while ranking of TT values were as follows: SAO-GO-WPIO > GO-WPIO > SAO-WPIO > GO-SAO > SA-G-WPI > G-WPI = G-SA> SA-WPI. The lowest PR and TT values were determined for SA-WPI (20.5 ± 0.12 N) and G-WPI (21.9 ± 0.12 N), respectively. The overall assessment of laminate films is presented in Figure 1. As shown, the laminated film SAO-GO-WPIO possessed the best mechanical properties, followed by GO-SAO, GO-WPIO, SAO-WPIO, SA-G-WPI, G-SA, G-WPI and SA-WPI (Table 2).

Mechanical characteristics for single-layer films manufactured from WPI, G and SA were reported previously and displayed in Table 3 [16]. The polysaccharide-based single layer film (SA) showed the highest tensile strength (15.3 ± 0.42 MPa); while the protein-based, single-layer film (G) produced the highest elongation (45.3% ± 3.60%). In all investigated laminated films, the entities containing polysaccharides (SA), in general, had the highest tensile strengths, which indicated that polysaccharides contributed to the durability of laminated films more than proteins (G and WPI). This result suggested that even stronger edible films could be made by laminating extra layers of polysaccharides-based films. In contrast, protein-based films (WPI and G) possessed better elastic properties than polysaccharides-based films (SA), as shown in Table 3. EB values of laminated films were higher than those of single polysaccharides-based SA films and lower than protein-based WPI and G films (Figure 1). The properties of PR and TT for laminated films were dramatically enhanced by lamination.

### 2.3. Film Thickness and Barrier Properties

The thickness of laminated films varied from 47.4 ± 2.76 (G-SA) to 112.3 ± 15.69 µm (SAO-GO-WPIO) as shown in Table 4. All laminated films manufactured with the addition of O were thicker than corresponding films manufactured without the addition of O. Thickness values for all films assessed were significantly different (*p* < 0.05) from each other, with the exception of that for GO-SAO and SAO-WPIO films. This occurred for the latter two laminated films because they were composed of very similar ingredients. Ranking of laminate film thickness decreased in the order: SAO-GO-WPIO > SA-G-WPI > GO-WPIO > G-WPI > GO-SAO = SAO-WPIO > SA-WPI > G-SA. In general, thinner films were more desirable in appearance and were less bulky.

Water vapour permeability (WVP) and oxygen permeability (OP) of laminated films showed that all tested laminated films (G-SA, G-WPI, SA-WPI, SA-G-WPI, GO-SAO, GO-WPIO, SAO-WPIO, GO-SAO and SAO-GO-WPIO) were significantly different (*p* < 0.05) to each other, while GO-SAO films showed the best barrier properties to water vapour (22.6 ± 4.04 g mm/kPa d m^2^) and oxygen (18.2 ± 8.70 cm^3^ µm/m^2^ d kPa), Table 4. In contrast, G-WPI showed the higher WVP (78.8 ± 13.12 g mm/kPa d m^2^), and SAO-GO-WPIO the higher OP (52.5 ± 11.45 cm^3^ µm/m^2^ d kPa). Barrier properties and films thickness are represented in Figure 2.

Ranking of WVP values increased in the order: GO-SAO > SAO-WPIO > SAO-GO-WPIO > SA-WPI > G-SA > GO-WPIO > SA-G-WPI > G-WPI; while OP values increased in the order: GO-SAO > SA-WPI > G-WPI > SAO-WPIO > GO-WPIO > G-SA > SA-G-WPI > SAO-GO-WPIO.

It is interesting to note that monolayered films for WPI had much higher WVP and OP values of 138.3 ± 15.49 g mm/kPa d m^2^ and 199.0 ± 4.00 cm^3^ µm/m^2^ d kPa, respectively, and as shown in Table 5.

Significant increases (*p* < 0.05) in barrier properties to water vapour and oxygen were observed for GO-SAO films in comparison to single-layer films formed from either G or SA (Table 4 and Table 5). This result suggested that the addition of lipid, such as O, followed by a pH adjustment of the film forming solution modified the biopolymer chemical structure such that the barrier properties of the resulting films improved [20]. It was reported that the WVP for gelatin films decreased, as chain length and concentration of fatty acids increased [1,20]. Similarly, [1] studied the WVP of bilayer films consisting of stearic and palmitic acids as one layer and HPMC as the other under various conditions of temperature and relative humidity. The films were expected to perform well at relative humidity below 90% and temperatures from −19 to 40 °C. Regarding OP, the addition of O in laminated structures did not decrease OP dramatically when compared with samples that did not contain O, with the exception of GO-SAO samples. One of the other possible reason of the WVP and OP difference in laminated structures is film thickness variety.

The lamination process represents an important step toward the engineering of protein films for packaging application. Although, WVP values for laminated edible/biodegradable/compostable films still have higher WVP values compared to those of commercial synthetic polymer films, such as low density polyethylene ((LDPE), WVP of 0.00385–0.00582 g mm/ m^2^ d kPa) and high density polyethylene ((HDPE), WVP of 0.000987–0.00237 g mm/ m^2^ d kPa) [23]; WVP values were converted for comparison with edible films by a [24] permeability calculator. These results confirmed that lamination of components differing in physical properties is a viable method for film property improvement.

Moreover in the addition to WVP and OP barrier properties UV light barrier property is essential for some types of food in order to prevent intramolecular rearrangements under UV light, which can turn edible substances in packed food into non edible or even harmful. WPI films were shown to have excellent barrier properties to UV light, regardless of composition in the region between 200 and 280 nm [25]. Thus, represented laminates structures containing WPI layers could have UV protective properties.

### 2.4. Microstructure of Laminated Films

In an attempt to elucidate the films structural characteristics that are of importance in terms of understanding mechanical film properties and films resistance to gas transmission, SEM was used to visualise the surface of duplex and triplex laminated films (SA-G WPI-G, SAO-GO and SA-G-WPI).

Film cross-sections of laminated films are shown in Figure 3a–d. As can be seen, micrographs indicated that the laminated layers were tightly bound with all base layer films due to surface interaction or adhesion, with the exception of that for G-WPI, which is probably a consequence of negative charges on the surfaces between WPI and G. Throughout this study, all manufactured laminated films were found to retain their integrity and stability.

A fairly smooth and orientated SA layer and a compact G layer, in which corn oil was incorporated, can be observed in Figure 3b (GO-SAO). This was a very different structural appearance from that of the G-SA laminated film (Figure 3a). The G-SA film showed a rough and porous cross-sectional character. The gelatin layer in the GA-SA laminated film has roughly the same structure as that of a single layer G film, as was shown previously by SAM [20].

Smoothness of film texture is evident in SAM images as the addition of O increases to create an emulsion by the inclusion of lipid globules into the gelatin matrix [20]. In Figure 3b, monolayer G and GO films structures are compared to G-SA laminated film. As can be seen again, and similar to the GO layer in the laminate structure, a smooth GO monolayer structure can be observed [20]. GO-SAO films demonstrated the best mechanical properties among all of the laminated films tested, indicated by possessing the highest TS, PR and TT values, and lowest WVP and OP values. GO-SAO films compared to G-SA films showed that WVP value decreased by 60%, and OP value by 54%, which is also in agreement with [20].

SA-G-WPI was the most elastic laminated film among the films tested in this study. Two protein layers (G and WPI) played a predominant role in terms of providing elasticity. Three-layer components (SA, G and WPI) can be clearly distinguished from the micrograph (Figure 3c). The cross-section of G and WPI layers display ridges and valleys, suggesting a ductile specimen. However, SA-G-WPI films also had the second highest OP values, which is a most undesirable feature when applied as a food packaging material. Due to proteins being hydrophilic by nature, G-WPI films had the highest WVP values. It has been evident in this study that the use of optimized formulations was necessary to produce laminated films, in order to impart edible/biodegradable/compostable films with stronger physical properties and lower transmission rates to gas.

One of the common issues for compostable films that can affect the film structure is its biodegradation on the top surface. It is well known that polymer films composed of proteins and polysaccharides are attractive substrates for different microorganisms, especially bacteria and fungi. Consequently, it will be necessary to investigate the addition of specific antimicrobials to the polymer composition, which were shown to prolong the shelf life of packed food such as sodium octanoate or bitter oranges extracts in gelatin films [26], aminobenzoic or sorbic acids, sodium lactate (NaL) and 3-polylysine in whey protein films [27,28], and many others antimicrobials systems [29]; and which are often required to prevent microbial growth on the films’ surface. The other proposed laminates’ protection can be developed by the application of a very thin wrapping film on the top side in order to prevent moisture loss from the products, laminates structures and fungi growth.

## 3. Conclusions

Laminated films were successfully produced by employing proteins (G and WPI), polysaccharide (SA) and the addition of oil (O) producing films GO, WPIO and SAO. Most of the tensile strength, puncture resistance, elongation at break and tear strength, as well as water vapour permeability, oxygen permeability and thickness of laminated films were significantly different from each other and from those of single layer films (G, WPI or SA). In comparison to the single layer films, the TS, PR and TT properties of laminated films were enhanced considerably, especially for laminated films produced using optimal formulations with the addition of O. However, elasticity of laminated films was lower than that of single layer films. This is a negative attribute for film material in packaging applications. Although WVP and OP values of laminated films were not dramatically decreased in comparison to single layer films (G or SA), with the exception of that for GO-SAO film, it is still a very interesting approach to improve overall properties and functionality of edible/biodegradable/compostable films by laminating different film layers of various compositions. The laminate that distinguished itself among all others in terms of general performance was GO-SAO. This specific laminate warrants further investigation for food packaging applications. For example, it will be reasonable to protect films from microbial and fungi growth on the top surface by the incorporation of different antimicrobials or to use laminates in addition to a tiny layer of a protecting film.

## 4. Materials and Methods

### 4.1. Ingredients for Films Formation

Gelatin (Bloom 180) was purchased from Redbook Ingredient Services Ltd., Dublin, Ireland; whey protein isolate (Bipro, protein > 97.8%) was purchased from Davisco Foods International INC, MN, USA; sodium alginate was purchased from Manugel DMB, ISP Ltd., Surrey, UK; glycerol was delivered from Cahill May Roberts Ltd., Dublin, Ireland; pure corn oil was delivered from Mazola, produced for Best Foods UK Ltd., Esher Surrey, UK; NaOH was purchased from Lab Pak Ltd., Filongley, UK and lactic acid was delivered from VWR International, Alkem Chemicals Ltd., Cork, Ireland.

### 4.2. Preparation of Film Forming Solutions and Film Formation

WPI (20 g), G (20 g) and SA (5 g) were separately solubilised in distilled water to obtain solutions of WPI (4 wt %), G (4 wt %) and SA (1 wt %) with desirable concentrations to form films. The addition of glycerol to each solution was set to glycerol/powder ratio of 1:2 (*w:w*). Corn oil (O) containing solutions to form optimal GO, WPIO and SAO films were prepared by adding corn oil and pH adjustment using lactic acid or 1 M NaOH before heating the solutions. All solutions were stirred continuously using a magnetic stirrer hotplate until powders were completely dissolved. Solutions were homogenised at 480 bar (first stage at 30, second stage at 450) using APV homogeniser 2000 series (APV, Alberslund, Denmark) three-times after heating to 80 °C. The base films were casted by pouring solutions onto levelled Teflon-coated Perspex plates and dried for 24 h at 50% ± 5% RH and 23 ± 2 °C. The laminate layers were poured directly onto the base films and then dried for up to 72 h. The volume ratio and the components used in the construction of the multilayer films are shown in Table 1. Formed films were subsequently peeled from the casting plates and held under the same conditions for a further 12 h prior to testing.

### 4.3. Film Thickness

Film thickness was measured using a 0–25 mm screw gauge (Mitutoyo Corporation, Kawasaki, Kanagawa, Japan) with overall thickness being expressed as an average (*n* = 15) taken randomly from each film. Film thickness was used in calculating TS, WVP and OP values.

### 4.4. Mechanical Properties

Mechanical properties of films were evaluated according to the ASTM-D882 [30] standard test methodology using an Imperial 2500 instrument, Mecmesin force and torque test solutions (Mecmesin Ltd., Slinfold, West Sussex, England). Test film samples were cut into strips (100 mm × 25.4 mm) and analysed for TS, EB, TR and PT.

### 4.5. Films WVP

Circular water vapour permeability cups made from Perspex were manufactured to the specifications reported by [31]. Briefly, distilled water (6 mL) was placed in each test cup and a film sample was mounted across the cup opening. The cups were stored under controlled temperature and humidity (50% ± 3% RH, 23 ± 2 °C). A constant air velocity of 152 m min^−1^ was maintained over the cups to ensure uniform air movement across the WVP test cells. Steady state conditions were reached within 2 h. The weight loss of the cups was monitored over a 24 h period with weights recorded at 2 h intervals. Water vapour permeability was calculated according to the protocol specifications, which is a modification of the ASTM E-96 standard method [32] for determining WVP of synthetic packaging materials.

### 4.6. Films OP

The measurement of OP was conducted at controlled condition (50% ± 3% RH, 23 ± 2 °C) according to the method developed by [33]. Film was mounted between the upper lid and rubber ring with silicon lubricant and fixed to the lower cup of the reported fixture, with an oxygen sensor housed inside. Nitrogen gas was blown into the chamber through one pipe, while exhausting through the other until the nitrogen reading becomes stable within the chamber. Both pipes were then shut. The sensor measured the declining nitrogen content over time. The data was graphed, and the developed equation was used to calculate the profile phase.

### 4.7. Scanning Electron Microscopy

Scanning electron microscopy (SEM) was performed on a JSM-5510 (SEM, JEOL Ltd. Tokyo, Japan) at 5 kV. Film samples were examined for cross-section characteristics, which were affixed to aluminium stubs with double-sided cellophane tape and sputter-coated with a layer of gold prior to imaging.

### 4.8. Statistical Analysis

Measurements for TS, EB, PR, TR, TT, WVP and OP were performed on 6 replicates. A significance of 95% confidence level by a Duncan’s multiple range test was used for all statistical analysis.

## Figures and Tables

**Figure 1 ijms-21-02486-f001:**
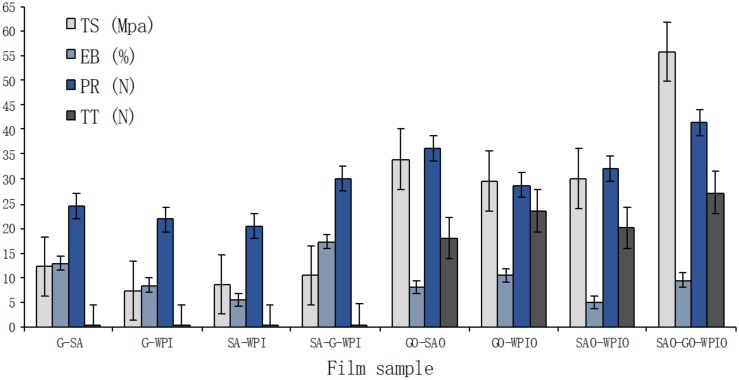
Values for tensile strength (TS), elongation at break point (EB), puncture resistance (PR) and tear strength (TT) of laminated films. Means ± standard deviation for *n* = 6.

**Figure 2 ijms-21-02486-f002:**
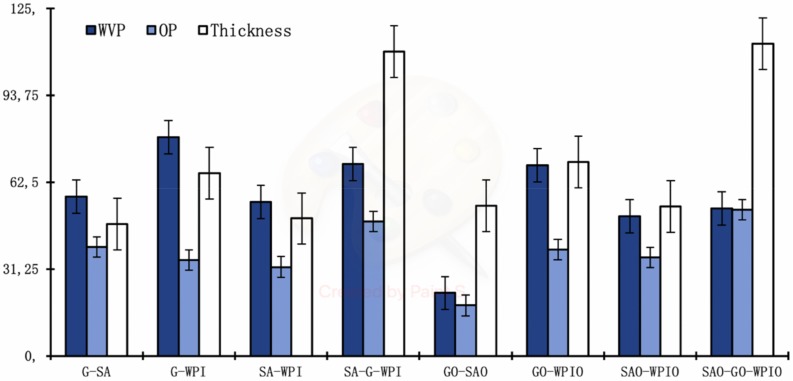
Values of WVP (g mm/m2 d kPa), OP (cm^3^ µm/m^2^ d kPa) and the thickness (µm) of laminated films. Means ± standard deviation for *n* = 6.

**Figure 3 ijms-21-02486-f003:**
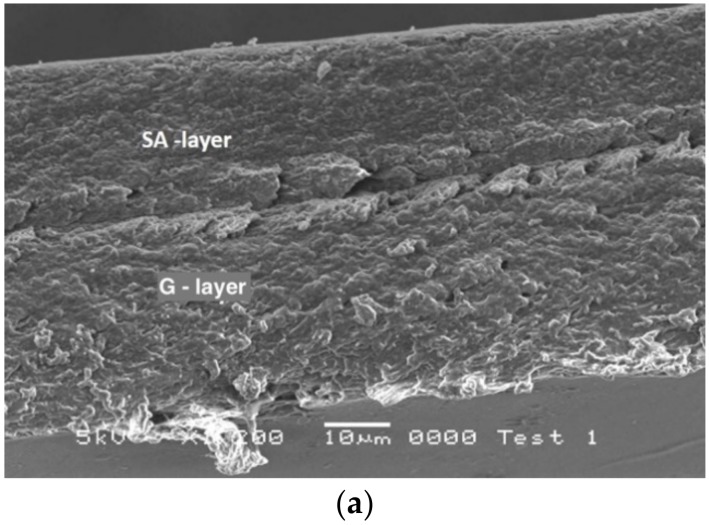
(**a**) Cross-section of G-SA laminated film and (**b**) cross-section comparison of GO-SAO laminated film (on the left, this study), G monolayer film and GO film with 27.25% corn oil (on the right) [20]. (**c**) Cross-section of SA-G-WPI laminated film. (**d**) Cross-section of G-WPI laminated film.

**Table 1 ijms-21-02486-t001:** Volume ratio and components of the investigated laminate films.

Films	Base Layer (mL)	First Laminating Layer (mL)	Second Laminating Layer (mL)	Dry Mass Ratio
Laminated films manufactured using optimised films without corn oil				
G-SA	G 20	SA 70	-	0.8:0.7
G-WPI	G 20	WPI 20	-	0.8:0.8
SA-WPI	SA 70	WPI 20	-	0.7:0.8
SA-G-WPI	SA 70	G 30	WPI 30	0.7:1.2: 1.2
Laminated films manufactured using optimised films with corn oil				
GO-SAO	GO 20	SAO 70	-	0.8:0.7
GO-WPIO	GO 20	WPIO 20	-	0.8:0.8
SAO-WPIO	SAO 70	WPIO 20	-	0.7:0.8
SAO-GO-WPIO	SAO 70	GO 30	WPIO 30	0.7:1.2: 1.2

G: gelatin; WPI: whey protein isolate; SA: sodium alginate; GO: gelatin film forming solution was prepared at its optimal formulation with the addition of corn oil; SAO: sodium alginate film forming solution was prepared at its optimal formulation with the addition of corn oil; WPIO: whey protein isolate film forming solution was prepared at its optimal formulation with the addition of corn oil; G-SA: G films laminated with SA films; G-WPI: G films laminated with WPI films; SA-G-WPI: SA films laminated with G films, then to WPI films; GO-SAO: GO films laminated with SAO films; GO-WPIO: GO films laminated with WPIO films; SAO-WPIO: GO films laminated with WPIO films; SAO-GO-WPIO: SAO films laminated with GO films, then with WPIO films.

**Table 2 ijms-21-02486-t002:** Mechanical properties of laminated films ^a^.

Film Samples	TS (MPa)	EB (%)	PR (N)	TT (N)
G-SA	12.3 ± 0.04 ^d^	12.9 ± 0.03 ^b^	24.6 ± 0.06 ^f^	0.2 ± 0.09 ^f^
G-WPI	7.4 ± 0.01 ^g^	8.5 ± 0.02 ^d^	21.9 ± 0.12 ^g^	0.2 ± 0.13 ^f^
SA-WPI	8.7 ± 0.11 ^f^	5.1 ± 0.07 ^f^	20.5 ± 0.12 ^h^	0.2 ± 0.08 ^g^
SA-G-WPI	10.4 ± 0.01 ^e^	17.4 ± 0.03 ^a^	30.2 ± 7.02 ^d^	0.4 ± 0.07 ^e^
GO-SAO	34.6 ± 0.13 ^b^	8.1 ± 0.05 ^e^	36.2 ± 0.07 ^b^	18.1 ± 0.06 ^d^
GO-WPIO	29.5 ± 0.66 ^c^	10.1 ± 0.72 ^c^	28.9 ± 1.56 ^e^	23.7 ± 0.53 ^b^
SAO-WPIO	30.1 ± 4.43 ^c^	5.0 ± 0.78 ^g^	32.1 ± 2.91 ^c^	20.2 ± 5.09 ^c^
SAO-GO-WPIO	55.8 ± 7.98 ^a^	9.5 ± 1.55 ^c^	41.4 ± 10.01 ^a^	27.3 ± 3.45 ^a^

^a^ Means ± standard deviation for *n* = 6. Any two means in the same column followed by the same letter ^(a–h)^ are not significantly (*p* > 0.05) different as determined by a Duncan’s multiple range test.

**Table 3 ijms-21-02486-t003:** Mechanical properties of edible films formed from single ingredients [16] ^a^.

Film Samples	TS (MPa)	EB (%)	PR (N)	TT (N)
WPI, 8%	5.3 ± 0.54 ^a^	22.5 ± 6.61 ^a^	9.9 ± 1.08 ^a^	0.14 ± 0.049 ^a^
G, 4%	5.7 ± 0.02 ^a^	45.3 ± 3.6 ^b^	10.1 ± 1.27 ^a^	0.13 ± 0.039 ^b^
SA, 1%	15.3 ± 0.42 ^b^	4.7 ± 0.83 ^c^	13.9 ± 0.77 ^b^	0.01 ± 0.000 ^c^

^a^ Means ± standard deviation for *n* = 6. Any two means in the same column followed by the same letter ^(a–c)^ are not significantly (*p* > 0.05) different as determined by a Duncan’s multiple range test.

**Table 4 ijms-21-02486-t004:** Water vapour permeability and oxygen permeability of laminated films ^a^.

Film Samples	WVP (g mm/kPa d m^2^)	OP (cm^3^ µm/m^2^ d kPa)	Thickness (µm)
G-SA	57.3 ± 11.03 ^d^	39.2 ± 6.05 ^c^	47.4 ± 2.76 ^g^
G-WPI	78.8 ± 13.12 ^a^	34.5 ± 8.18 ^f^	65.7 ± 6.89 ^d^
SA-WPI	55.45 ± 11.11 ^e^	32.0 ± 9.02 ^g^	49.4 ± 4.38 ^g^
SA-G-WPI	69.1 ± 8.06 ^b^	48.3 ± 12.46 ^b^	109.4 ± 12.52 ^b^
GO-SAO	22.6 ± 4.04 ^h^	18.2 ± 8.70 ^h^	54.0 ± 4.49 ^e^
GO-WPIO	68.6 ± 17.32 ^c^	38.3 ± 7.09 ^d^	69.8 ± 8.98 ^c^
SAO-WPIO	50.3 ± 12.67 ^g^	35.4 ± 16.67 ^e^	53.7 ± 5.84 ^e^
SAO-GO-WPIO	53.2 ± 5.16 ^f^	52.5 ± 11.45 ^a^	112.3 ± 15.69 ^a^

^a^ Means ± standard deviation for *n* = 6 for water vapour permeability (WVP) and oxygen permeability (OP); *n*= 24 for thickness. Any two means in the same column followed by the same letter ^(a–h)^ are not significantly (*p* > 0.05) different as determined by a Duncan’s multiple range test.

**Table 5 ijms-21-02486-t005:** Water vapour permeability and oxygen permeability of single layer films [16] *.

Film Sample	WVP (g mm/ m^2^ d kPa)	OP (cm^3^ µm/m^2^ d kPa)	Thickness (µm)
WPI, 8%	138	90	95.7 ± 0.07
G, 4%	56	32	58.3 ± 0.05
SA, 1%	24	18	25.9 ± 0.04

* Standard deviations for WVP and OP are omitted.

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
