# Peer review of "Development and Assessment of Duplex and Triplex Laminated Edible Films Using Whey Protein Isolate, Gelatin and Sodium Alginate"

_ijms, 2020, doi:10.3390/ijms21072486_

Round 1

Reviewer 1 Report

With respect to the great interest in production and use of edible/biodegrad–able/compostable films especially in food packaging this paper is well written based on sound research and thus delivers a lot of interesting and high-valued data. 

I had only two suggestions: according to the flow of reading and to numbering of Tables chapters 2 "Results and Discussion" and chapter 3 "Materials and Methods" should change place - or the Tables have to be renumbered, at least (when following the instructions for authors).

The references should be numered and listed according to their appearance in text instead of alphabetical order in accordance with the instructions for authors.

Another suggestion was to add data of additional films already in use for comparison e.g. cellophane and/or polyesters like PET, PBT or PBS. This would enhance the comparabilty with existing packaging solutions, although the later ones are not at all or or only to a very limited extend edible/compostable/biodegradable. This would be a nice extension of this paper.

Author Response

Dear Reviewer,

Thank you for your time spent for the reviewing of our manuscript. We very appreciate your notes and high mark for our research.

Please, find below the following answers to your questions/notes:

I had only two suggestions: according to the flow of reading and to numbering of Tables chapters 2 "Results and Discussion" and chapter 3 "Materials and Methods" should change place - or the Tables have to be renumbered, at least (when following the instructions for authors)

- Sections 2 and 3 were changed.

The references should be numbered and listed according to their appearance in text instead of alphabetical order in accordance with the instructions for authors.

- Reference numbers were changed according to the manual and numbered, please, see in the text. Reorder was made.

Another suggestion was to add data of additional films already in use for comparison e.g. cellophane and/or polyesters like PET, PBT or PBS. This would enhance the comparabilty with existing packaging solutions, although the later ones are not at all or or only to a very limited extend edible/compostable/biodegradable. This would be a nice extension of this paper.

- Thank you for this suggestion. I would make the paper better, but there are some issues exist with the data comparison for these materials. Usually oxygen transmission and water vapour transmission rates are compared to each other within a batch of films, but they depend on testing method applied, film thickness and RH. So only when materials tested at the same conditions, testing method and % of RH barrier properties can be compared to each other. Those materials such as PET, BPS are usually tested by MOCON method (or IR detecting method) and it will be hard to find similar film thickness and RH in the literature in order to compare with our films. In our case we used different method for WVP - cup method and for OP we used our own developed method sited. So barrier properties can't be compared even with similar barrier properties of PET or PBS due to different methods were applied. This statement we are going to include in our new Review for gelatin films barrier properties comparison depending on different methods applied and formulations. The review we are going to submit soon in this May.

Regarding the comparison with oil sourced materials, the following comparison with conventional oil sourced films was added: "The lamination process represents an important step toward the engineering of protein films for packaging application. Although, WVP values for laminated edible/biodegradable/compostable films still have higher WVP values compared to those of commercial synthetic polymer films, such as low density polyethylene (LDPE), WVP of 0.00385 – 0.00582 g mm/ m2 d kPa and high density polyethylene (HDPE), WVP of 0.000987 – 0.00237 g mm/ m2 d kPa) [29]; WVP values converted for comparison with edible films by [30] permeability calculator). These results confirmed that lamination of components differing in physical properties is a viable method for film property improvement."

At the same time knowing the troubles with the films data comparison we have added Tables 4 and 5 where we added OP and WVP of single layered films from which our laminates consist of and which we studied at similar conditions in our previous paper sited in: 

Wang L.Z., Auty Mark A.E., Rau A., Kerry J.F., Kerry J.P. Effect of pH and addition of corn oil on the properties of gelatin-based biopolymer films. Journal of Food Engineering, 2009, 90 (1), 11-19.

Reviewer 2 Report

1. The introduction is rather brief and lacks information about the ecological significance of the development of biodegradable packaging materials and coatings from biopolymers, as well as about the biological aspects of the film application and further degradation. For example, it is well known that polymer films composed of proteins and polysaccharides are attractive substrates for different microorganisms, especially bacteria and fungi. Addition of specific stabilizers to the polymer composition is often required to prevent microbial growth. The authors should discuss this important issue in details and provide additional references to prove that application of such biodegradable films does not significantly decrease the shelf life of the products packed.   2. The assignment of the polymer layers in the electron micrographs is not quite evident for the readers (especially in Fig. 3c), so it would be better to include also the SEM images of the cross-sections for the monolayer films studied (i.e. WPI, G and SA) for clarity.   3. In the experimental section the accelerating voltage for SEM studies is given as 10 kV, while the micrographs in Fig. 3a-3d clearly indicate 5 kV. Please, revise the SEM experiment description carefully. It is also recommended to describe the procedure of the cross-sections' preparation in details.  

Author Response

Thank you very much for your time for careful consideration of our manuscript and for your valuable notes and mistakes found. Please, find the following respond to your questions.

1a. The introduction is rather brief and lacks information about the ecological significance of the development of biodegradable packaging materials and coatings from biopolymers.

- We absolutely agree with your offer to include ecological aspects of biopolymers. The additional introduction paragraph was added:

"Conventional packaging materials, which have included the use of wood, paper-based products, glass, metals (primarily steel and aluminium) and plastics, plastic-based laminates and co-extrusions have formed the cornerstone to those formats which have, and still for the most-part, underpin the commercial packaging used around retailed products and throughout all three packaging levels, primary (sales), secondary (collation and handling) and tertiary (transport). These materials have been successful workhorses and have been responsible to a major degree for; the packaging processes and technologies that exist today, the distribution and transportation systems responsible for moving goods successfully all over the world and the manner in which all products are displayed and merchandised today. Additionally, the packaging materials and formats employed have led to the reduction of product costs (especially for food), wide and almost constant availability of product types regardless of seasonality, creation of competition in the marketplace, and provision of consumer convenience in many different forms. This is particularly true for food and beverage products, where features of product safety, hygiene and shelf-life stability have become important as features of convenience, particularly through packagings role in reducing food waste. When discussing the problems associated with food packaging materials on environmental grounds, it is extremely important not to lose sight of the role played by packaging to date in reducing food wastage, surely an issue as important as packaging waste, if not more so. Therefore, while much controversy surrounds the use of current, conventional, non-biodegradable packaging materials such as plastics and plastic-based packaging on environmental grounds, it is important that we do not exert knee-jerkreactions which result in adopting packaging materials and formats which are not robust enough to successfully contain, protect or preserve food products. This will only result in creating further negative environmental issues. Packaging pollution of our global environment has been in the making since the 1950s and the real failure associated with packaging is not the packaging waste itself, but the human failure in the first instance to put systems in place to deal with different packaging materials when initially introduced into the marketplace for use."

1b. as well as about the biological aspects of the film application and further degradation. For example, it is well known that polymer films composed of proteins and polysaccharides are attractive substrates for different microorganisms, especially bacteria and fungi. Addition of specific stabilizers to the polymer composition is often required to prevent microbial growth. The authors should discuss this important issue in details and provide additional references to prove that application of such biodegradable films does not significantly decrease the shelf life of the products packed.

- In section 3.4 we added the following paragraph: "One of the common issue for compostable films which can effect the film structure is its biodegradation on the top surface. It is well known that polymer films composed of proteins and polysaccharides are attractive substrates for different microorganisms, especially bacteria and fungi. Consequently, it will be necessary to investigate the addition of specific antimicrobials to the polymer composition which were shown to prolong the shelf life of packed food such as sodium octanoate or bitter oranges extracts in gelatin films [32], aminobenzoic or sorbic acids, sodium lactate (NaL) and 3-polylysine in whey protein films [33, 34], and many others antimicrobials systems [35]; and which are often required to prevent microbial growth on the films’ surface. The other proposed laminates’ protection can  be developed by the application of a very thin wrapping film on the top side in order to prevent moisture loss from the products, laminates structures and fungi growth."

Also, we added this statement in Conclusion: 

"This specific laminate warrants further investigation for food packaging applications. For example, it will be reasonable to protect films from microbial and fungi growth on the top surface by the incorporation of different antimicrobials or to use laminates in addition to a tiny layer of a protecting film."

2. The assignment of the polymer layers in the electron micrographs is not quite evident for the readers (especially in Fig. 3c), so it would be better to include also the SEM images of the cross-sections for the monolayer films studied (i.e. WPI, G and SA) for clarity.

We added G monolayer film SEM cross-sections with the addition of corn oil and without (on Figure 3b. Cross-section comparison of GO-SAO laminated film (on the left, this study), G monolayer film and GO film with 27.25% corn oil (on the right), [31]).

The only one example for WPI monolayer film was found in FANG Y., TUNG M.A., BRITT I.J., YADA S., DALGLEISH D.G. Tensile and Barrier Properties of Edible Films Made from Whey Proteins. J.  Food Science, 2002, 67(1), 188-193, figure 8. The cross section of this film (Figure 8c) revealed a sponge- like structure where whey-protein aggregates appear to be linked by fine strands to form a continuous network. But this picture looks different than our texture on images for WVP cross-section film in laminate structure for GO-SAO.

Fang et al., 2002 represented on Fig. 8c-d WPI monolayers cross-sections of the WVP films contenting quite similar glycerol content (40 agains ours 50%), but with the addition of Calcium chloride salt. In our study we didnt add any salts. It is known Ca2+ can bind neighbour groups in polymer chain and thus can effect to a film structure. Thus, we decided no to add this example due to we haven’t used the addition of salts.

In addition, studying the paper of Fang et al., 2002 is was revealed that WPI films has an excellent UV barrier regardless film composition. We have added the other paragraph in section 3.3:

"Moreover in the addition to WVP and OP barrier properties UV light barrier property is essential for some type of food in order to prevent intramolecular rearrangements under UV light which can turn edible substances in packed food into non edible or even harmful. WPI films were shown to have excellent barrier properties to UV light, regardless of composition in the region between 200- to 280-nm [31]. Thus, represented laminates structures containing WPI layers could have UV protective properties."

3. In the experimental section the accelerating voltage for SEM studies is given as 10 kV, while the micrographs in Fig. 3a-3d clearly indicate 5 kV. Please, revise the SEM experiment description carefully. It is also recommended to describe the procedure of the cross-sections' preparation in details.  

Voltage was changed to 5 kV in Materials and methods. Our microscopist said that was enough for SEM images description.